# Demographic and Pathogens of Domestic, Free-Roaming Pets and the Implications for Wild Carnivores and Human Health in the San Luis Region of Costa Rica

**DOI:** 10.3390/vetsci8040065

**Published:** 2021-04-20

**Authors:** Joseph Conrad, Jason Norman, Amalia Rodriguez, Patricia M. Dennis, Randall Arguedas, Carlos Jimenez, Jenifer G. Hope, Michael J. Yabsley, Sonia M. Hernandez

**Affiliations:** 1College of Veterinary Medicine, University of Georgia, Athens, GA 30602, USA; jcon77@gmail.com (J.C.); jasonnormandvm@gmail.com (J.N.); 2Former Costa Rica Campus, University of Georgia, San Luis 11604, Costa Rica; mjyabsley1@gmail.com; 3Department of Veterinary Preventive Medicine, College of Veterinary Medicine, The Ohio State University, Columbus, OH 43210, USA; dennis.129@osu.edu; 4Cleveland Metroparks Zoo, Cleveland, OH 44109, USA; 5Zoologico Simon Bolivar, San Jose 10101, Costa Rica; corytophanes@yahoo.com; 6Escuela de Medicina Veterinaria, Universidad Nacional de Costa Rica, Heredia 40101, Costa Rica; carlos.jimenez.sanchez@una.ac.cr; 7HOPE Animal Medical Center, Athens, GA 30606, USA; Jenivet@aol.com; 8Warnell School of Forestry and Natural Resources, University of Georgia, 180 E Green Street, Athens, GA 30602, USA; myabsley@uga.edu; 9Southeastern Cooperative Wildlife Disease Study, College of Veterinary Medicine, University of Georgia, Athens, GA 30602, USA

**Keywords:** conservation, parasites, vector-borne pathogens, wildlife, zoonoses

## Abstract

Habitat loss and degradation, restricted ranges, prey exploitation, and poaching are important factors for the decline of several wild carnivore populations and additional stress from infectious agents is an increasing concern. Given the rapid growth of human populations in some regions like Costa Rica, pathogens introduced, sustained, and transmitted by domestic carnivores may be particularly important. To better understand the significance of domestic carnivore pathogens for wildlife, we determine the prevalence of infection and possible mechanisms for contact between the two groups. The demographics, role in the household, and pathogens of pet dogs and cats was studied during three annual spay/neuter clinics in San Luis, Costa Rica. Most dogs were owned primarily as pets and guard animals, but ~10% were used for hunting. Cats were owned primarily as pets and for pest control. Both roamed freely outdoors. We detected high prevalences of some pathogens (e.g., carnivore protoparvovirus 1 and *Toxoplasma gondii*). Some pathogens are known to persist in the environment, which increases the probability of exposure to wild carnivores. This study demonstrated that domestic pets in San Luis, home to a number of protected and endangered wildlife species, are infected with pathogens to which these wild species are potentially susceptible. Additionally, results from our questionnaire support the potential for domestic and wild animal contact, which may result in disease spillover.

## 1. Introduction

As a result of direct effects such as habitat loss and degradation, restricted ranges, prey exploitation, and poaching, several wild carnivore populations in Central America currently face declines [1]. The potential additional stress from infectious agents, introduced, sustained, and transmitted by domestic carnivores, could further contribute to this decline [2]. Habitat loss and/or fragmentation and hunting may also contribute to an increase in contact among domestic and wild carnivores. Even when direct contact is not obvious, domestic carnivores may contribute to pathogen presence and persistence in the ecosystem. Understanding the presence and prevalence of these pathogens, and the risk of introduction and persistence of relevant pathogens in sensitive areas, should be a priority for conservation.

Domestic dogs and cats serve as reservoirs for pathogens to which wild canids and felids are susceptible [1,2,3,4,5,6] and spillover from domestic animals to wild populations has been demonstrated repeatedly. Domestic dogs have been suspected or proven to be reservoirs for disease outbreaks of wild carnivore populations, including rabies and canine adenovirus (CAV) in Ethiopian wolves (*Canis simensis*) and canine distemper virus (CDV) in African wild dogs (*Lycaon pictus*), Serengeti lions (*Panthera leo*), and spotted hyenas (*Crocuta crocuta*) [7,8]. There is ample evidence that wild felids, canids, mustelids, procyonids, and marsupials have been infected by pathogens originating from domestic carnivore populations [1,9,10,11,12,13,14,15,16,17,18,19,20,21,22,23,24,25]. These spillover events have occurred even in areas far from urban centers with large populations of domestic animals, such as Bolivia’s Kaa-Iya del Gran Chaco and Noel Kempff Mercado National Parks [18,20]. Domestic dogs and cats can also serve as sentinels to detect the circulation of important pathogens, including some that are of zoonotic concern [26,27,28,29,30,31,32]. Currently, there is little understanding of the pathogens that pose a risk for wild carnivores in Central America, or whether domestic animals play a role in these risks. In addition to understanding what pathogens are being maintained by domestic animals, it is important to know what amount of contact these animals may have with wildlife (e.g., are animals allowed to free roam, do they hunt with owners, are they vaccinated/treated for parasites, etc.).

The San Luis valley of Costa Rica is a unique ecosystem composed of a small town surrounded by a matrix of dairy farms, small coffee plantations, and forest fragments. It is bordered by the three large private reserves that have made the Monteverde region of Costa Rica a premier ecotourism destination in Central America. This region is home to numerous wild felids, canids, mustelids, mephitids, procyonids, and marsupial species (Table 1), all which could be considered at risk for pathogen spillover events from domestic carnivores. All five relevant felid species are listed as Appendix I by the Convention on International Trade in Endangered Species of Wild Fauna and Flora (CITES) and are protected throughout Costa Rica. Although listed as species of least concern at the population level, the greater grison (*Galictis vittata*), the cacomistle (*Bassariscus sumichrasti*), and the olingo (*Bassaricyon gabbii*) occur at low densities, are considered uncommon or rare, and, specifically in Costa Rica, are listed as endangered (CITES Appendix III; [33,34]). Tayras (*Eira barbara*) are one of the most common medium-sized wild predators observed in the San Luis region, and the authors have encountered them frequently on farms where they likely to come into contact with domestic pets. Striped hog-nosed skunks (*Conepatus semistriatus*) have also adapted to human disturbance and occur on the edges of coffee plantations. Neotropical otters (*Lutra longicaudis*) are found in the San Luis river (which borders the town of San Luis), yet their population status is data deficient. Although not afforded any special protection, white-nosed coatis (*Nasua narica*) and kinkajous (*Potus flavus*) are reported to be decreasing throughout their range. Kinkajous are common visitors to coffee plantations and farms where they take advantage of continuously fruiting trees. Although there is no documented evidence of pathogens causing disease outbreaks in Central American marsupials, agents such as *Leptospira* and *T. gondii* have been documented in other species of opossums and the potential of spillover exists [9,16].

In addition to the risk posed to native fauna, human populations are susceptible to a number of zoonotic diseases for which domestic animals serve as reservoirs, including numerous bacterial (*Brucella canis*, *Leptopira interrogans* spp.), viral (rabies virus), fungal (*Microsporum canis*, *Sporothrix schenckii*), and parasitic (*Toxocara canis*, *Toxocara cati*, *Toxoplasma gondii*) diseases, either through direct contact with infected pets or indirectly through contact with contaminated food or water sources [35,36,37]. The most common disorders are gastrointestinal (salmonellosis, parasitic), dermatologic (dermatophytoses, scabies, cutaneous larval migrans, erythema migrans), respiratory (psittacosis), and multisystemic (rabies, leptospirosis, toxoplasmosis, toxocariasis) diseases, all of which contribute to considerable economic loss due to medical treatment and inability to work and also play a major role in local attitudes and treatment of free-ranging domestic, feral, and wild animal populations [38].

The primary objective of this study was to determine the demographics, the role in the household, and the exposure to and infection with selected pathogens, of rural domestic pet dogs and cats in a conservation-sensitive region of Costa Rica.

## 2. Materials and Methods

### 2.1. Study Site

This study took place in the San Luis valley, located in the northwest region of Costa Rica approximately 7 km from the town of Santa Elena in the Monteverde region, in the province of Guanacaste. The town of San Luis is composed of two sections (“upper” and “lower” San Luis) along an altitudinal gradient. At 1100 m, the former University of Georgia (UGA) San Luis Research Station (10 16′57,117″ N 84 47′53,747″ W) is located in “upper San Luis” and served as the base for the described work. Although the Monteverde region gained worldwide acclaim for its large private preserves, surrounding areas such as the San Luis valley are in strong contrast to these preserves [39,40]. These surrounding areas are more typical of modern tropical landscapes, composed of many small forest fragments and shade-grown coffee parcels within a matrix of pasture for livestock. The San Luis valley is classified as premontane wet forest with a temperature ranging from 15 to 22 °C and an average rainfall of 300 cm.

### 2.2. Biological Sample Collection and Storage

From July 2007 to July 2009, biological samples were obtained from domestic carnivores during three preventive medicine and annual sterilization clinics. A community liaison (AR), worked extensively to estimate the number of pets in the town, advertise the preventive medicine clinic each year, and obtain further details. A questionnaire was administered by native Spanish speakers to collect details about the origin of the pet, its role in the household, medical history of the pet, level of nutrition, previous preventive medicine, relationship with other domestic animals, roaming status, and potential for contact with wildlife. A complete and detailed physical exam was performed on each animal prior to anesthesia, particularly to determine if the animal had any preexisting condition that would preclude it from undergoing surgery. All animals underwent general anesthesia and surgical sterilization. Immediately thereafter, 1–3 mL of blood, feces, and ectoparasites were collected. A blood smear was immediately made, air dried, stained with a modified Wright–Giemsa stain (Diff-Quik^®^, Fisher Scientific Company LLC, Kalamazoo, MI, US, and stored at room temperature. The remaining blood volume was equally divided between a blood collection tube with EDTA (BD Microtainer Tubes, BD, Franklin Lakes, NJ, USA) and a serum separator tube (BD Serum Separator Tubes, BD, Franklin Lakes, NJ, USA). Blood samples were centrifuged at 10,000 rpm for 10 min and serum was pipetted and stored in sterile Eppendorf tubes (Micro Centrifuge Tubes, VWR International, West Chester, PA, USA) at −20 °C until transport. Fecal samples were collected via direct digital palpation of the rectum, were divided into equal halves and stored in formalin and 2.5% potassium dichromate. All samples were exported to the USA for further processing.

### 2.3. Serologic Assays

Domestic cats were tested for *Dirofilaria immitis* and feline leukemia virus (FeLV) antigens and feline immunodeficiency virus (FIV) antibodies (SNAP^®^ Combo FeLV Ag/FIV Ab test or SNAP^®^ 3Dx™, IDEXX Laboratories, Westbrook, MA, USA), following the manufacturer’s recommendations. In addition, cats were tested for the presence of antibodies against *Toxoplasma gondii* at the University of Georgia’s Athens Diagnostic Laboratory (UGAADL). Canine blood samples were tested for the presence of *D. immitis* antigen and antibodies against *E. canis*, *A. phagocytophilum*/*A. platys*, and *Borrelia burgdorferi*, according to the manufacturer’s recommendations (SNAP^®^ 4Dx^®^ Test, IDEXX Laboratories, Westbrook, MA 04092, USA). In addition, serum samples from dogs were tested for antibodies against canine distemper virus (CDV), carnivore protoparvovirus 1 (canine parvovirus), *Brucella canis*, and six *Leptospira interrogans* serovars (*L. bratislava*, *L. canicola*, *L. grippotyphosa*, *L. hardjo*, *L. icterohaemorrhagiae*, and *L. pomona*) at the UGAADL. See Table 2 for details.

### 2.4. Molecular Assays

DNA was extracted from blood samples using a commercial DNA extraction kit (DNeasy Blood and Tissue Extraction Kit, QIAGEN, Germantown, MD, USA). DNA isolated from canine blood samples collected in 2008 and 2009 was tested by polymerase chain reaction for *Babesia* spp., *Hepatozoon* spp., *Ehrlichia canis*, and *Anaplasma* spp. as previously described [41]. DNA from feline blood samples collected in 2008 and 2009 was tested for *Cytauxzoon* spp., *Babesia* spp., and *Anaplasma phagocytophilum* as previously described [41,42]. Positive amplicons were bidirectionally sequenced at the Integrated Biotechnology Laboratories (University of Georgia, Athens, GA, USA) to confirm identity. Chromatogram data were analyzed using Sequencher (Ann Arbor, MI, USA). To prevent and detect contamination, DNA extraction, primary and secondary amplification, and product analysis were done in separate dedicated areas. A negative water control was included in each set of DNA extraction, and a different water control was included in each set of primary and secondary PCR reactions.

### 2.5. Analysis of Blood Films and Fecal Samples

Blood smears were examined for the presence of hemoparasites following standard procedures [43]. Fecal samples stored in 2.5% potassium dichromate solution were examined microscopically using a direct fecal smear technique. Fecal samples stored in formalin were examined using the fecal flotation technique, utilizing sodium nitrate as the flotation solution (Feca-Med, First Priority Inc., Elgin, IL, USA).

### 2.6. Ethical Statement

This study was conducted according to the guidelines of the Declaration of Helsinki and was approved by and conducted under the guidelines and requirements of the Colegio de Medicos Veterinarios de Costa Rica (College of Veterinary Doctors of Costa Rica). In addition, this was part of a community low-cost spay and neuter clinic that was facilitated by community liaisons and local veterinarians. Informed consent was obtained from all subjects that brought their pet dogs or cats to the clinic.

## 3. Results

### 3.1. Demographics and Role of Pets in the Household and Ecosystem

The town of San Luis is composed of approximately 80 households of which approximately 70% reported to have at least one domestic dog or cat in the household (Amalia Rodriguez, pers comm.). We examined 64 dogs (37 female, 27 male; 39 adult, 25 juvenile) and 41 cats (20 female, 21 male; 22 adult, 19 juvenile). The results of questionnaires over the three-year study period indicate that the majority of dogs were owned primarily as pets (69%) compared to guard animals (17%) while 15% served both purposes. In addition, 10% of the dogs were used for hunting. Only a small minority of dogs lived primarily indoors (3%), with the large majority living outdoors (81%) and some considered them both indoor and outdoor pets (16%), but regardless, 100% were allowed to free roam outside at some point during the day (Table 3). Cats were owned primarily as pets (42%), for pest control (27%), or both (32%). Again, only a small percentage was allowed to live primarily indoors (10%); although a larger proportion of cats, when compared to dogs, were allowed to live both in/outdoors (36%) (Table 4). All cats living outdoors were allowed to free roam.

Only 25% of dogs received some vaccination, although most people could not describe what kind, and 36% were treated with an unidentified antiparasitic drug at some time in their lifetime. Only one dog (2%) had ever been examined by a veterinarian (Table 5). As expected, cats received more sparse preventive medical care; none were vaccinated and only 20% were given antiparasitic drugs at some point in their life (Table 5). Of all the animals examined in this study, one dog was reported by the owner to not be eating well at the time of exam, but all other dogs and cats were reported to be in overall good health. The majority of animals (80% of dogs; 87% of cats) came from households with multiple pets.

### 3.2. Serologic and Molecular Assays

A significant number of dogs had antibodies against CPV (63%), whereas the prevalence of antibodies against CDV was low (14%) (Table 6). Of the three dogs with antibodies against *L. interrogans*, antibodies were found against *L. canicola, L. grippotyphosa,* and *L. icterohaemorrhagiae*. Antibodies to *E. canis* and *Anaplasma* spp. were detected in 15% and 2% of dogs, respectively (Table 6). All dogs were negative for heartworm antigen and antibodies to *Brucella canis* and *B. burgorderi*. In 2008 and 2009, a total of seven (21%) dogs were PCR positive for *Hepatozoon canis*, one (3%) dog was PCR positive for *E. canis*, and one (6%) was PCR positive for *A. platys*.

Antigen of FeLV was detected in only one cat (from 2007) (5%) and all cats were negative for antibodies to FIV and *B. burgdorferi* and antigens of *D. immitis*. Antibodies to *T. gondii* were detected in 14 (42%) cats (Table 7), titers ranged from 1:32 to 1:512 (five 1:32, six 1:64, one 1:256, and two 1:512). PCR testing of cats indicated that two (13%) were positive for *E. canis* (both in 2008). All cats from 2008 and 2009 were PCR negative for *Cytauxzoon, Babesia*, and *Hepatozoon*.

### 3.3. Parasitology

We collected a total of 40 fecal samples from dogs, of which 65% had at least one species of intestinal parasite. *Ancylostoma caninum* (hookworm) was the most prevalent (53%) (Table 8). We obtained 21 fecal samples from cats, of which 24% contained at least one parasite. The prevalence of all three parasites found in cats (*Toxocara cati*, *Ancylostoma tubaeformae*, and *Isospora* spp.) was equal (10%) (Table 9).

## 4. Discussion

The transmission of domestic carnivore diseases to wild carnivore populations has become an issue of increasing concern in conservation [44]. However, in order for disease transmission from domestic animals to wild populations to occur, the following criteria must be met: (1) the wild population must be susceptible to the disease, (2) the disease must be present in the domestic animal population, and (3) there must be contact between domestic and wild populations or the pathogen in question must persist in the environment. That requires that researchers demonstrate (1) the presence of pathogens in the domestic animal population, (2) susceptibility of wild carnivores to pathogens found in domestic carnivores, and (3) some mechanisms by which domestic carnivores are contributing to the pathogen load in their shared ecosystem. Our study supports the presence of pathogens in domestic carnivores, and previous research has established the susceptibility of relevant wild carnivores in the San Luis region to these pathogens [1,2,5,9,10,11,12,13,14,15,16,17,18,19,20,21,22,23,24]. Other than applying radio telemetry and video surveillance techniques to domestic free-roaming pets, it is extremely difficult to confirm and quantify the direct contact they have with wild carnivores. To the best of our abilities, the results of our questionnaire provide evidence that there is the potential for contact between domestic and wild carnivores.

Epidemiological surveys of viral diseases in domestic dog and cat populations have been conducted in many developed countries but relatively few reports detailing their prevalence have been conducted in tropical countries, although a worldwide distribution of the viruses in this study is suspected. Although we have found no report detailing the country-wide prevalence of canine distemper in Costa Rica, case reports in dogs exist and conversations with local veterinarians confirm that, as with other Central American countries, it is considered a disease of major importance for domestic dogs [45]. In regard to wildlife health, recently, infection has been recognized in wild felids and clinical disease was reported in coatis [46,47]. Given the lack of preventive medicine in this region, we expected the prevalence of antibodies against canine distemper to be higher. The most likely explanation for such a low prevalence is that CDV has a high mortality rate and thus, it is likely that dogs that acquired the infection died and were not available for sampling at the time of our study. It is also possible that the circulation of CDV would be higher in unowned animals and surrounding wildlife species and there is only sporadic transmission to owned dogs in the region. Our study area is considered rural and a related study in Chile found that the prevalence of CDV was highest in domestic dogs and red fox (*Vulpes vulpes*) in urban environments compared to rural areas, likely due to higher densities and population sizes [48,49]. Canine parvovirus emerged in the mid-1970s as a new pathogen of dogs and has since become endemic in the global dog population. This virus is now recognized as a genogroup of the species carnivore protoparvovirus 1 which now also includes feline panleukopenia virus. Parvoviruses are extremely stable in the environment. Indirect transmission likely plays an important role in the transmission and maintenance of viruses in a population, particularly in wild carnivore populations in this region, where their density is low. Transmission between domestic and wild carnivores also occurs readily, and while direct transmission through close contact or predation has been proposed, the viruses are also readily transmitted by fomites or mechanical vectors such as insects, rodents, or vehicles wherever they are present in the ecosystem [41]. Infection with or exposure to carnivore protoparvovirus 1 has been reported in New World canids [50,51,52,53]. The prevalence of carnivore protoparvovirus 1 in this population of domestic dogs was significant (63%) and we would expect activities such as free roaming to contribute to the dissemination of this virus into the environment, especially since dogs can transport the virus on their hair coats.

Exposure of free-ranging felids to FeLV, FIV, and CDV has been established in several countries [1]. There is little information on the status of infection or exposure to FeLV, FIV, and CDV in free-ranging felids in Latin America, although Avendaño et al. [46] recently reported infection of free-ranging ocelots and pumas with CDV. These viruses have caused infection and mortality linked to pathogen spillover from domestic cats to closely related felids in many countries and so are a conservation concern [54,55,56,57]. Selected viral diseases of domestic cats have primarily been surveyed in metropolitan regions of Costa Rica [58] and thus our report is significant in a rural region where contact among domestic and wild felids is more likely. Interestingly, the prevalence of FeLV and FIV in our study was very low, likely due to a relatively low density of cats in the San Luis region.

Little is known about the prevalence of intestinal parasites of dogs and cats in Costa Rica, although the prevalence in dogs ranges from 5–40% depending on the location, age, and owned status [59]. A similar survey conducted in 2009 in San Isidro, Costa Rica found similar prevalence rates in dogs with *T. vulpis* (27%)*,* dogs and cats with *Ancylostoma* spp. (75% and 1%, respectively), and dog and cats with *Toxocara* spp. (5% and 1%, respectively) [60]. The prevalence of *T. cati* in cats from other countries in Latin America has been reported to be much higher (26–70%), although studies in cats are limited in geographic scope and many are older [60]. Similar prevalences (7–53%) were reported for *T. canis* from dogs from the same countries, demonstrating that these intestinal parasites are widespread in Central America [61]. Specifically, the common canine intestinal parasites, *Toxocara* and *Ancylostoma*, have been reported from fecal and soil samples from several areas in Costa Rica [62,63].

Heartworm is a serious parasite of dogs and cats that has a wide geographic range. The presence and prevalence of the parasite depends primarily on the density of appropriate mosquito vectors and susceptible canid hosts. Felids, ursids, pinnipeds, and humans are accidental hosts and rarely contribute to the maintenance of the parasite. There have been several studies investigating the prevalence of *D. immitis* in Costa Rica in dogs and although the prevalence varied geographically, it was generally low at 4–22% and several sites had no infections (e.g., Tibas and Santa Ana which are all in the same province and at a similar elevation to San Luis) [60,64,65,66]. The lack of heartworm in San Luis is likely due to a lack of appropriate vectors or no history of infected animals in the region.

Three of the tick-borne pathogens we detected in dogs (*E. canis, A. platys*, and *H. canis*) utilize a common vector, *Rhipicephalus sanguineus.* These three organisms are common canine pathogens in areas where the vector is present, and a high prevalence of *E. canis* has been reported in dogs from Costa Rica, especially in clinically ill dogs [60,64,67,68,69]. In fact, the dog that was PCR positive for *E. canis* in the current study was a juvenile with clinical signs suggestive of clinical ehrlichiosis, including splenomegaly, decreased clotting time, and a low percent cell volume. When questioned after surgery, the owner admitted the dog had a history of lethargy. The brown dog tick, *R. sanguineus*, is widely distributed throughout the world and has been reported repeatedly from wild mammals in Latin American countries [70,71]. *E. canis* and *A. platys* have recently been detected in wild and/or domestic felines and *E. canis* infections in cats can be associated with anemia, thrombocytopenia, lymphopenia, and monocytosis [72,73,74,75,76], although there are several incidental reports of *E. canis* in wild felids and domestic cats from Brazil, Trinidad and Tobago, and the United States [72,75,76,77,78]. Similarly, reports of *A. platys* in domestic cats from Brazil and Chile were from apparently healthy cats [6,73,79]. To the best of our knowledge, our report is the first of *E. canis* in a domestic cat in Central America. We did not detect *B. canis vogeli* in any dogs, which was unexpected, since it has been reported from Costa Rica, utilizes the same vector as *E. canis, A. platys*, and *H. canis*, and often occurs in a similar prevalence to *H. canis* [41,80,81,82]. Although we found no ticks on cats included in this study, *R. sanguineus* was common on dogs, and we likely failed to detect the tick on cats because it is a rare ectoparasite of cats and we only examined a limited number of cats during a brief period of the year. The status of the effects of these tick-borne pathogens on individual or wild animal populations remains unknown; however, we do know that the density of their vectors is highly dependent on the presence of domestic animals, without which these pathogens would not exist.

The presence of zoonotic parasites (e.g., *Toxocara*, *Ancylostoma, Leptospira*) in the domestic animal population also poses a possible public health concern. For example, human exposure to *Toxocara* in Costa Rica is not uncommon [83] and ascarid and hookworm eggs have been reported in public parks and beaches in Costa Rica [61]. Interestingly, human dirofilariasis cases tend to be rare, but there are at least six reports in Costa Rica in lowland regions where canine heartworm is more prevalent [84,85,86]. Should global climate predictions for the Monteverde region become a reality, the vector for *D. immitis* could expand into this region and become established [87]. The region of San Luis, situated near Costa Rica’s second most visited touristic region, is growing rapidly. The rise in the human population and the subsequent increase in domestic animals will further increase the interaction between these two groups. The town of San Luis does not have a local doctor and residents must often travel into Santa Elena to see a health care worker or to one of the larger cities (>3 h away) for adequate medical care, making this community vulnerable to emergent diseases.

There were several limitations to this study. Sample collection in this study was performed on animals voluntarily presented for a sterilization clinic, which may not accurately reflect the actual population of pet animals in the area. For example, owners who took advantage of the spay/neuter clinic may be more inclined to actively pursue health care and disease prevention for their pets than owners who did not attend. Community liaisons cited owner neglect, lack of education, inability to capture or handle the pet, and transport difficulties as reasons for why some owners did not bring their pets to the sterilization clinics. As an added incentive, animals that were sterilized were also vaccinated. In 2007 and 2008, all dogs were vaccinated against distemper, hepatitis, leptospirosis, parainfluenza, carnivore protoparvovirus 1 (DHLPP), and rabies, and all cats were vaccinated for feline viral rhinotracheitis, calicivirus, panleukopenia (FVRCP), and rabies. In 2009, all dogs and cats received rabies vaccinations. Stray animals, although rare, do exist in San Luis, but they were not included in this study. These animals almost certainly do not receive even the most basic preventive medicine or health care and thus the prevalence of those pathogens included in this study may differ or these animals may harbor different pathogens we did not detect. In addition, annual sample sizes, especially for intestinal parasites, were generally small. The sample size for cats was also much smaller than that for dogs, but cats are less commonly kept as pets and are less likely to receive veterinary care compared with dogs. Thus, the absence of certain parasites (e.g., ascarids, whipworms, etc.) could be due to small samples sizes of each age class.

In developed nations, regular visits to a veterinarian for preventive medicine have proven to be an effective method for the control of disease in domestic animal populations [88]. This is reasonably feasible in most communities and has reduced the risk of disease in domestic animals, their owners, and local wildlife [88]. However, in developing nations, particularly in rural areas where economic stresses and reduced access to proper veterinary care are a concern, preventive medicine is not practiced, which can lead to frequent turnover and endemic transmission of pathogens [89]. In these cases, government-run, mass vaccination programs and public education about proper hygiene and disposal of pet waste may be the more appropriate tools for controlling disease. Public education has been shown to be an effective means for controlling zoonotic disease, and mass vaccination programs for diseases such as rabies and canine distemper reduced infection rates in human and wild animal populations, respectively [38]. Furthermore, decreasing contact between wild and domestic animal populations through the constraint of domestic animals in yards or houses might help to reduce the potential for disease transmission between these populations. This could be accomplished through political intervention with legislation governing the restriction of animal movement. However, this may not be practical in this area, as the free-roaming lifestyle of the dogs and cats of the San Luis area is a direct consequence of their role in the family. Dogs are used for protection, and are still used for hunting, although the results of our survey likely underestimate this activity, as people in San Luis understand this is illegal. One of the authors has personally witnessed hunting parties with dogs in search of paca (*Agouti paca*). Dogs and cats are often used for pest control around the home, which dictates a free-roaming lifestyle. In this case, public education about the potential threat of disease could help to outweigh the perceived benefit of allowing pets to roam freely and compliance with new laws may be more reasonable. Future investigations should focus on disease surveillance of endangered wild animals in Costa Rica. Baseline information regarding pathogen exposure in these species is critical for their population health and future survival.

## 5. Conclusions

This study demonstrates that domestic pets in the San Luis region of Costa Rica, home to a number of protected and endangered wildlife species, have either been exposed to or are actively infected with a number of pathogens to which these wildlife species are potentially susceptible. These pathogens include FeLV, *T. gondii*, *E. canis*, *H. canis*, L. *interrogans*, CDV, CPV, *T. cati*, *A. tubaeformae*, *A. caninum*, *T. canis*, *Isospora* spp., and *Trichuris* spp. The potential for agent spillover from domestic to wild animal populations exists and serves to add to the severe survival pressures already faced by these populations due to human activity.

## Figures and Tables

**Table 1 vetsci-08-00065-t001:** Wild mesomammals susceptible to pathogens of domestic carnivores of San Luis, Costa Rica.

Felidae	Canidae	Mustelidae	Mephitidae	Procyonidae	Marsupialidae
Jaguar (*Panthera onca*)	Coyote (*Canis latrans*)	Long-tailed weasel (*Mustela frenata*)	Striped hog-nosed skunk (*Conepatus semistriatus*)	Cacomistle (*Bassariscus sumichrasti*)	Common opossum (*Didelphis marsupialis*)
Puma (*Felis concolor*)	Gray fox (*Urocyon cinereoargenteus*)	Greater grison (*Galictis vittata*)		Raccoon (*Procyon lotor*)	Central American wooly opossum (*Caluromys derbianus*)
Ocelot (*Leopardus pardalis*)		Tayra (*Eira barbara*)		White-nosed coati (*Nasua narica*)	Common Gray four-eyed opossum (*Philander opossum*)
Margay (*Leopardus weidii*)		Neotropical river otter (*Lutra longicaudis*)		Kinkajou (*Potus flavus*)	Water opossum (*Chironectes minimus*)
Jaguarundi (*Puma yaguarondi*)				Olingo (*Bassaricyon gabbii*)	Alston’s mouse opossum (*Micoureus alstoni*)
					Mexican mouse opossum (*Marmosa mexicana*)

**Table 2 vetsci-08-00065-t002:** Methodologies and positive cut-off values used to detect exposure to selected disease agents of dogs and cats from San Luis, Costa Rica.

Pathogen	Methodology	Positive Cut-Off
Canine distemper virus	Antibody SN ^a^	1:32
Carnivore protoparvovirus 1	Antibody HI ^b^	1:20
*Dirofilaria immitis*	Antigen bidirectional flow ELISA ^c^	P/N ^e^
Feline immunodeficiency virus	Antibody bidirectional flow ELISA	P/N
Feline leukemia virus	Antigen bidirectional flow ELISA	P/N
*Anaplasma* spp.	Antibody bidirectional flow ELISA	P/N
*Ehrlichia canis*	Antibody bidirectional flow ELISA	P/N
*Borrelia burgdorferi*	Antibody bidirectional flow ELISA	P/N
*Leptospira interrogans*	Antibody MA ^d^	1:100
*Toxoplasma gondii—feline*	Antibody ELISA	1:32

^a^ SN: serum neutralization. ^b^ HI: hemagglutination inhibition. ^c^ ELISA: enzyme-linked immunosorbent assay. ^d^ MAT: micro-agglutination. ^e^ P/N: test scored as positive or negative.

**Table 3 vetsci-08-00065-t003:** Characteristics of domestic dogs from San Luis, Costa Rica.

Year	Owned as Pet	Owned as Guard Dog	Used for Hunting	Live Indoors Only	Live Outdoors Only	Live Indoors/Outdoors
2007 *n* = 29	16	7	3	1	24	4
2008 *n* = 17	15	2	2	0	12	5
2009 *n* = 18	13	2	1	1	16	1
Total (%)	44 (69)	11 (17)	6 (10)	2 (3)	52 (81)	10 (16)

**Table 4 vetsci-08-00065-t004:** Characteristics of domestic cats from San Luis, Costa Rica.

Year	Owned as Pet	Owned for Pest Control	Owned as Pet and for Pest Control	Live Indoors Only	Live Outdoors Only	Live Indoors/Outdoors
2007 (*n* = 20)	8	4	8	4	13	3
2008 (*n* = 16)	7	5	4	0	4	12
2009 (*n* = 5)	2	2	1	0	4	1
Total (%)	17 (42)	11 (27)	13 (32)	4 (10)	21 (51)	16 (39)

**Table 5 vetsci-08-00065-t005:** Preventive medicine administered to domestic dogs and cats in San Luis, Costa Rica. No./No. interviewed (percent).

Year	Received Vaccination		Received Antiparasitic Agent		Examined by a Veterinarian	
	Dogs	Cats	Dogs	Cats	Dogs	Cats
2007	7/29 (24)	0/20	19/29 (66)	3/20 (15)	0/29	0/20
2008	3/17 (18)	0/16	8/17 (47)	5/16 (31)	0/17	0/16
2009	6/18 (33)	0/5	6/18 (33)	0/5	1/18 (6)	0/5
Total	16/64 (25)	0/41	23/64 (36)	8/41 (20)	1/64 (2)	0/41

**Table 6 vetsci-08-00065-t006:** Serologic results for selected pathogens of domestic dogs from San Luis, Costa Rica (2007-2009).

Sample Year	No. Positive/No. Tested (%)				
	Canine Distemper Virus (CDV)	Carnivore Protoparvovirus 1 (Canine Parvovirus, CPV)	*Leptospira* *interrogans*	*Ehrlichia* *canis*	*Anaplasma* spp.
2007	1/26 (4)	20/26 (77)	3/26 (12)	6/28 (21)	0/28
2008	3/16 (19)	9/16 (56)	0/16	1/16 (6)	0/16
2009	4/17 (24)	8/17 (47)	0/17	2/17 (12)	1/17 (6)
Total	8/59 (14)	37/59 (63)	3/59 (5)	9/61 (15)	1/61 (2)

**Table 7 vetsci-08-00065-t007:** Serologic results for selected pathogens of domestic cats from San Luis, Costa Rica, 2007–2009.

Sample Year	No. Positive/No. Tested (%)			
	Feline Leukemia Virus (FeLV)	Feline Immunodeficiency Virus (FIV)	*Toxoplasma gondii*	*Dirofilaria immitis*
2007	1/20 (5)	0/20	8/18 (44)	ND
2008	0/15	0/15	4/15 (27)	0/15
2009	0/5	0/5	2/5 (40)	ND
Total	1/40 (3)	0/40	14/38 (37)	0/15

ND. Not done.

**Table 8 vetsci-08-00065-t008:** Prevalence of intestinal parasites in dogs from San Luis, Costa Rica. Number Positive/No. Tested (%).

Sample Year	*Toxocara canis*	*Isospora* spp.	*Ancylostoma* spp.	*Trichuris* spp.	Any Parasite
2007	2 /15 (13)	1/15 (7)	8/15 (53)	3/15 (20)	11/15 (73)
2008	0/11	1/11 (9)	4/11 (36)	0	5/11 (45)
2009	2/14 (14)	0	9/14 (64)	0	10/14 (71)
Total	4/40 (10)	2/40 (5)	21/40 (53)	3/40 (8)	26/40 (65)

**Table 9 vetsci-08-00065-t009:** Prevalence of intestinal parasites in domestic cats from San Luis, Costa Rica. Number Positive/No. Tested (%).

Sample Year	*Toxocara cati*	*Isospora* spp.	*Ancylostoma* spp.	Any Parasite
2007	0/12	0/12	0/12	0/12
2008	2/8 (25)	2/8 (25)	1/8 (13)	4/8 (5)
2009	0/1	0/1	1/1 (100)	1/1 (100)
Total	2/21 (10)	2/21 (10)	2/21 (10)	5/21 (24)

## Data Availability

The data generated in this study are all available within this article. No unique genetic sequences were generated. If other information is needed, it can be obtained from the corresponding author.

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
