# Peer review of "Demographic and Pathogens of Domestic, Free-Roaming Pets and the Implications for Wild Carnivores and Human Health in the San Luis Region of Costa Rica"

_vetsci, 2021, doi:10.3390/vetsci8040065_

Round 1

Reviewer 1 Report

Title. Change by including ‘demographic’. Demographic and pathogens of domestic….(see below)  
Introduction   Too much details regarding the different wild species in the area, but less information is given regarding the utility of understanding the demography and the role in the households of domestic dogs and cats….why these data is needed to justify this questionnaire survey?? See Acosta-Jamett et al., 2010, Prev Vet Med as a guide. If there is an interest in the demography of cats and dogs this should be included in the title. Vaccination is important, also whether owners allow their animals to roam freely….etc.   P3L101-103: I doubt that with this study a prevalence to any pathogen could be determined (sampling is only carried out in clinics). So, please remove the word ‘prevalence’ in this paragraph and change (i.e. rate, percentage, etc) accordingly throughout the manuscript.    Methods Why a physical exam was performed? This helped to fulfill any objective? if so please add this to the aim…in the MS P4L127 says that physical exam was carried out for assessing physical condition previous to any surgery….is it that needed for fulfilling the objectives of this MS?   Results Serology….animals that were seropositive to CPV, CDV…etc….were not vaccinated??, if so, please include in methodology    Discussion P7L242. Please help to understand how this study provide evidence that there is potential for contact between domestic and wild carnivores??. As far as I can see at least this study gives a % of pets that allowed to roam freely….   P7L252-255. It’s the only reason?. why not that this is not a maintenance area for this pathogen and sporadic transmission is observed? See Acosta-Jamett et al., 2011, Vet Microbiol.   Include in introduction/discussion that dogs are used as sentinel species for different pathogens, including these of zoonotic interest  that could aid to justify this study (REF Cleaveland, S., Meslin, F.X., Breiman, R., 2006. Dogs can play useful role as sentinel hosts for disease. Nature 440, 605.)

Author Response

Reviewer 1

Title. Change by including ‘demographic’. Demographic and pathogens of domestic….(see below)  

RESPONSE: Added as suggested.

Introduction   Too much details regarding the different wild species in the area, but less information is given regarding the utility of understanding the demography and the role in the households of domestic dogs and cats….why these data is needed to justify this questionnaire survey?? See Acosta-Jamett et al., 2010, Prev Vet Med as a guide. If there is an interest in the demography of cats and dogs this should be included in the title. Vaccination is important, also whether owners allow their animals to roam freely….etc.  

RESPONSE: Thank you for this suggestion. We have added statements on why it is important to understand demographics and role, in addition to simply knowing what pathogens are present in these animals.

“In addition to understanding what pathogens are being maintained by domestic animals, it is important to know what amount of contact these animals may have with wildlife (e.g., are animals allowed to free-roam, do the hunt with owners, are they vaccinated/treated for parasites, etc.). “

P3L101-103: I doubt that with this study a prevalence to any pathogen could be determined (sampling is only carried out in clinics). So, please remove the word ‘prevalence’ in this paragraph and change (i.e. rate, percentage, etc) accordingly throughout the manuscript.   

 RESPONSE: Changed as suggested (e.g., in first instance, changed to just “…the exposure to and infection with…”. However, there were instances later in the manuscript where we retained that term as it was a prevalence of antibodies detected WITHIN that set of samples tested. We agree that this does not necessarily align with what the prevalence may be in a larger sample size of animals. We noted that limitation in the discussion as well (i.e., unowned pets or those not brought to the clinic may have additional pathogens or different rates of infection with those we tested).

“These animals almost certainly do not receive even the most basic preventive medicine or health care and thus the prevalence of those pathogens included in this study may differ or these animals may harbor different pathogens we did not detected.”

Methods Why a physical exam was performed? This helped to fulfill any objective? if so please add this to the aim…in the MS P4L127 says that physical exam was carried out for assessing physical condition previous to any surgery….is it that needed for fulfilling the objectives of this MS?  

RESPONSE: A physical exam is conducted as part of best-practices for any surgery to minimize risks. Although not a part of our study per-se, it is important to note this to illustrate humane animal care.

Results Serology….animals that were seropositive to CPV, CDV…etc….were not vaccinated??, if so, please include in methodology   

RESPONSE: This is certainly a valid concern with vaccinated animals. In this particular case we have retained our general results as they are for a few reasons. The history of vaccination was provided by the owners which is always subject to unknowns. First, only 25% of dog owners stated the dogs had any vaccine in the past. Most could not recall what vaccine their dog got or when they got the vaccine. In several instances it appeared they got first vaccines as puppies but then no more vaccinations. Thus, it is likely that many of these vaccinated dogs were not up-to-date.  Also, combo vaccines are rarely used and our understanding that most vaccinations were only for CDV. Interestingly, we only found a 14% prevalence of antibodies to CDV further supporting that at least some ‘vaccinated’ dogs did not have antibodies to CDV. 

Discussion P7L242. Please help to understand how this study provide evidence that there is potential for contact between domestic and wild carnivores??. As far as I can see at least this study gives a % of pets that allowed to roam freely….  

RESPONSE: Details on possible contact through roaming and hunting activities are provided in the last paragraph of the discussion.

P7L252-255. It’s the only reason?. why not that this is not a maintenance area for this pathogen and sporadic transmission is observed? See Acosta-Jamett et al., 2011, Vet Microbiol.  

RESPONSE: We have added additional discussion as well as points raised in Acosta-Jamett et al.

 Include in introduction/discussion that dogs are used as sentinel species for different pathogens, including these of zoonotic interest that could aid to justify this study (REF Cleaveland, S., Meslin, F.X., Breiman, R., 2006. Dogs can play useful role as sentinel hosts for disease. Nature 440, 605.)

RESPONSE: These sections have been expanded to include this concept. One that I actually do work with a lot (i.e., dogs as sentinels for tick-borne pathogens).

Reviewer 2 Report

This manuscript includes information about the results of a serological and molecular study evaluating different pathogens for cats and dogs in San Luis Region. There is limited information published about this subject in the area.

Minor comments:

-2.3 Serologic assays: Authors should include methods to detect pathogens using in-house techniques (Toxoplasma gondii, CDV, Carnivore protoparvovirus 1, Brucella canis, six Leptospira interrogans serovars)

-Table 2. Authors should correct that Feline immunodeficiency virus and feline leukemia virus is not and ELISA technique in this strict sense. Moreover, in relation to Toxoplasma gondii methodology, only ELISA instead of KELA.

-3.1. Lines 175 and 176, please, you should include what it is F, M, A, J.

-Line 204, you should eliminate presumably A. platys.

-Conclusion section. You should include the information about first report of E. canis in a cat in Central America

Author Response

Reviewer 2

This manuscript includes information about the results of a serological and molecular study evaluating different pathogens for cats and dogs in San Luis Region. There is limited information published about this subject in the area. RESPONSE: Thank you for your review.

Minor comments:

-2.3 Serologic assays: Authors should include methods to detect pathogens using in-house techniques (Toxoplasma gondii, CDV, Carnivore protoparvovirus 1, Brucella canis, six Leptospira interrogans serovars)

RESPONSE: These samples were submitted to a veterinary diagnostic lab which uses standard methods for the different tests used (i.e., serum neutralization, hemagglutination inhibition, ELISA, etc.). Additional details are not available from the lab.

-Table 2. Authors should correct that Feline immunodeficiency virus and feline leukemia virus is not and ELISA technique in this strict sense. Moreover, in relation to Toxoplasma gondii methodology, only ELISA instead of KELA.

RESPONSE: For clarity – we have changed the test description to bidirectional flow ELISA which is how IDEXX describes their tests. Toxoplasma test name updated.

-3.1. Lines 175 and 176, please, you should include what it is F, M, A, J.

RESPONSE: These have been expanded to - female, male; adult, juvenile

-Line 204, you should eliminate presumably A. platys.

RESPONSE: Changed to Anaplasma spp.

-Conclusion section. You should include the information about first report of E. canis in a cat in Central America

RESPONSE: This statement is in 6th paragraph of the discussion.

Reviewer 3 Report

Pathogens of Domestic, Free-roaming Pets and the Implications 2

for Wild Carnivores and Human Health in the San Luis Region 3 of Costa Rica.

This manuscript is verery well written and provides useful baseline information on the prevalence of pathogens in domesticated dogs and cats/potential for transmission to wildlife.

Excellent introduction and Discussion sections

well referenced

The main weakness is the relatively small number of animals examined per year.

DNA isolated in 2008-2009- how isolated/how stored? Collected from whom?  How come the results of the molecular assays are not presented in a Table?  Details of molecular tests? Controls?

The number of animals examined per year is modest- only 1 cat in 2009 was assessed for intestinal parasites.  Can meaningful conclusions be drawn from so small a sample size?

Author Response

Reviewer 3

This manuscript is very well written and provides useful baseline information on the prevalence of pathogens in domesticated dogs and cats/potential for transmission to wildlife.

Excellent introduction and Discussion sections

RESPONSE: Thank you for your review and comments.

well referenced

RESPONSE: Thank you for your review and comments.

The main weakness is the relatively small number of animals examined per year.

 RESPONSE: We agree with this assessment but were limited to who was brought the clinic each year.

DNA isolated in 2008-2009- how isolated/how stored? Collected from whom?  How come the results of the molecular assays are not presented in a Table?  Details of molecular tests? Controls?

RESPONSE: Additional details were added for extraction, storage, etc. The few results related to the molecular testing was included in the text vs. a table. Although details of the molecular tests (i.e., primers) were not added because they were the same as the cited protocols; however, details on controls were added.

The number of animals examined per year is modest- only 1 cat in 2009 was assessed for intestinal parasites.  Can meaningful conclusions be drawn from so small a sample size?

RESPONSE: We agree with this assessment but were limited to who was brought the clinic each year. And we do agree that not much can be concluded with data from cats but we wanted to present all data that were made available through these clinics. Hopefully these data will stimulate additional studies.